# Human-Dog Relationship during the First COVID-19 Lockdown in Italy

**DOI:** 10.3390/ani11082335

**Published:** 2021-08-07

**Authors:** Danila d’Angelo, Andrea Chirico, Luigi Sacchettino, Federica Manunta, Maurizio Martucci, Anna Cestaro, Luigi Avallone, Antonio Giordano, Francesca Ciani

**Affiliations:** 1Department of Veterinary Medicine and Animal Production, University of Naples Federico II, 80137 Naples, Italy; danila.dangelo@unina.it (D.d.); sacchettinoluigi@gmail.com (L.S.); cestaro.an@gmail.com (A.C.); avallone@unina.it (L.A.); ciani@unina.it (F.C.); 2Department of Psychology of Development and Socialization Processes, “Sapienza” University of Rome, 00185 Rome, Italy; 3Veterinary Behavior and Consulting Service at “Free Interdisciplinary Italian Dog Academy” (LAICI), 25126 Brescia, Italy; federica.manunta@yahoo.it; 4Azienda Sanitaria Locale 3, 10068 Piemonte, Italy; mauriziopsi@libero.it; 5Sbarro Institute for Cancer Research and Molecular Medicine and Center of Biotechnology, College of Science and Technology, Temple University, Philadelphia, PA 19122, USA; giordano@temple.edu; 6Department of Medical Biotechnologies, University of Siena, 53100 Siena, Italy

**Keywords:** dog, dog-human relationship, owner perception, dog behavioral symptoms, SARS-CoV2, COVID-19, welfare, wellbeing, validation Monash scale, Italy, lockdown, pandemic

## Abstract

**Simple Summary:**

The SARS-CoV2 pandemic forced an abrupt interruption of social contacts and interpersonal affective relationships all over the world, according to national directives. The SARS-CoV2 pandemic imposed an abrupt termination of social contacts and interpersonal affective relationships around the world for a period whose duration depended on national directives This situation caused considerable inconvenience with important repercussions also on the emotional state of people. The dog-human coevolutionary process has led to the development of common cognitive abilities and a marked sociability. In this study, performed through a national survey, we investigated the influence of the first lockdown in the human-dog relationship and to what extent the owners were able to perceive the discomfort of their dog. How much could the lockdown affect our dogs’ emotional states? How do owners rate their dogs’ quality of life during restrictions? How much did the affectivity and the perception of one’s dog as demanding or expensive before and during the lockdown vary in the human-dog relationship? These and other questions were proposed in the survey. The survey analysis highlighted a stability in the interaction and a decrease in the dogs’ costs perceived by the owners, considering the human-dog relationship. Furthermore, the evaluation of symptoms revealed a moderate percentage increase in anxious behavior in the dog, probably due to the change of routine. Overall, the human sample did not seem particularly worried about any discomfort experienced by their dog during quarantine.

**Abstract:**

The SARS-CoV2 pandemic forced an abrupt interruption of social contacts and interpersonal affective relationships all over the world, according to national directives. Many considerable inconveniences occurred with important repercussions also on the emotional state of people and their pets. We carried out a national survey to evaluate the human-dog relationship in a social isolation context using an adapted version of Monash Dog Owner Relationship Scale, the perception of the dogs’ discomfort by their human owners, and the resilience of the dog through the quantification of symptoms, in time of the first lockdown of the COVID-19 pandemic. The results highlighted that the human-dog interaction was similar during quarantine; however, there was lower owner’s perception of a dog’s cost during the quarantine than before it.

## 1. Introduction

Rigorous quarantine measures were undertaken by numerous countries, such as the United States, the United Kingdom, India, France, Germany, Italy, Spain, China, and Singapore, due to the COVID-19 pandemic [1]. It appears that the most effective strategies to slow the transmission of severe acute respiratory syndrome coronavirus 2 (SARS-CoV-2) are good hand hygiene, social distancing, and physical quarantine for suspected or confirmed cases [2]. Social isolation underlies symptoms such as anxiety, depression, fear, stress, and sleep problems. These same symptoms are frequently encountered during the COVID-19 pandemic [3,4]. Social isolation has been one of the issues that humans have had to face during this period.

The concept that “no man is an island” involves the idea that human beings feel a certain discomfort when they are isolated from others and need to be part of a community to satisfy the needs inherent to the human species [5]. Studies suggest that social isolation and loneliness are serious public health issues, especially among the elderly, and they are linked to increased mortality and morbidity [6].

The COVID-19 pandemic raised questions about the role that human-animal interactions play in the context of social distancing and isolation measures. Links between human mental health and well-being appear of particular interest at this time. According to a very recent study by Ratschen et al. [7], the detrimental psychological effects of COVID-19 lockdown could be mitigated by animal ownership.

It has been proposed that dogs influence mental well-being by modulating the human physiological response to stress, specifically via attenuation of the hypothalamic-pituitary-adrenal (HPA) axis and heart rate response to stressors [8,9]. Living with a dog may be associated with improved well-being for people with chronic pain [10]. The human-dog relationship is influenced by both canine and human characteristics [11]. We know dogs have learned the ability to recognize human emotions due to their domestication process, which has selected dogs with the best communication skills with humans [12]. Dogs can recognize the emotional expressions of human faces [13,14] and integrate bimodal sensory information to discriminate positive and negative emotions both in dogs and humans and this skill appears to exceed the ability of other animals [15].

During the pandemic period, the Italian Government—with the “#IoRestoaCasa Decree” (Decree of the President of the Council of Ministers of 9 March 2020) [16]—established the obligation for everybody to stay in their homes, the limitation of the free movement of people, and the closure of almost all work activities from 11 March-3 May 2020. Pet owners could only take short walks with dogs near their home, and it was forbidden to take them to parks and socialize with the other dogs or people. In such a specific context, where both social and work habits were stopped, the Social Exchange Theory offers a theoretical framework able to understand the dog-owner relationship. The Social Exchange Theory is a well- established psychological theory specifying that relationships are maintained only when there is a balance between perceived costs and benefits or when the perceived benefits outweigh the perceived costs. This theory was first proposed to apply to human-companion animal relationships by Netting and collaborators [17]. Based on this theory, some scholars developed the Monash Dog-Owner Relationship Scale (MDORS). This validated psychometric measure assesses the extent to which owners and dogs engage in shared activities, the perceived emotional closeness of the relationship, and the perceived costs of the relationship for the dog owner [18].

The aim of the present study was two-fold. Firstly, we investigated the human-dog relationship in a social isolation context using an adapted version of MDORS. Secondly, we evaluated the perception of the dogs’ discomforts by their human owners, in a time of the COVID-19 pandemic. Furthermore, we evaluated, through the quantification of symptoms, the resilience of the dog during the lockdown, forced to cope with a change in habits and interactions with its family.

## 2. Materials and Methods

Data were collected via an online survey written in the Italian language and administered between 8 and 18 May 2020, the time that corresponded to the Italian lockdown period. Participants were recruited using online and social advertisements. All participants were informed regarding the general purpose of the study and their rights to anonymity. Researchers collected participants’ written, informed consent before participating in the study. The time needed to complete the survey was approximately 10 min. Collected data were coded and processed anonymously. The survey relied on the information recollected from the participants invited to answer the questions comparing the pre-quarantine and quarantine periods.

### 2.1. General and Demographic Information on Dog Owners and Their Dogs

In this group, 20 items were placed, one of which proposed another question in the case of a positive answer. The items concerned information on the owner and composition of the family unit. Another group of items, on the other hand, concerned information relating to the dog and its management, including provenance and evidence of any behavioral problems.

### 2.2. Measures

Adapted versions of study measures were developed specifically for the target variables, according with veterinary literature dealing with the evaluation of dog symptoms and adapting the Monash Dog-Owner Relationship Scale (MDORS). The Italian version of the measures was translated from the English version by two English-Italian bilinguals using standardized back-translation procedures [19].

The MDORS is a 28-item, five-point, Likert multidimensional scale to measure the owners’ perceived relationship with their dog. The MDORS includes three subscales related to separate dimensions of the human-dog relationship: Owner-Dog Interaction, Perceived Emotional Closeness, and Perceived Costs. There are no existing data to determine the level of relationship quality (e.g., high, medium, or low) from MDORS scores. Therefore, scores can only be compared within a specified group of human-dog dyads. Higher scores in any of the three subscales of the MDORS indicate a positive perception with respect to that subscale, even if those variables belong to the subscale of perceived costs of MDORS. Higher scores in the Interaction level subscale of MDORS mean higher level of interaction. Higher scores in the Perceived Emotional Closeness subscale of MDORS mean higher emotional closeness, and higher scores in the Perceived Costs subscale of MDORS mean lower perceived costs for the owner. For the purpose of the present study, we used only four items of the Owner-Dog Interaction and six items of the Perceived Costs subscales (please see the survey used in the study, at: https://osf.io/5m6wb accessed on 6 August 2021). The items were selected both considering original factor loadings and the context of quarantine, during which it was prohibited to go out for any activities (e.g., going to visit friends and familiars).

### 2.3. Information on Dog’s Lifestyle during the Quarantine

This section was made up of 15 items, two of which proposed another question in the case of a positive answer. The items concerned any changes in the dog’s lifestyle during the lockdown. In particular, changes in the composition of the family unit, possible removal of the dog from the home, and changes in the walk routine, also with respect to intraspecific sociability.

### 2.4. Information on Clinical and Behavioral Symptoms in the Dogs during the Quarantine

This section was made up of 16 items, two of which proposed another question in the case of a positive answer. Eleven items predicted a response in relation to the intensity of clinical and behavioral symptoms compared to the period prior to the lockdown. Participants answered using a Likert-like scale with 5 points: much less than before, less than before, equal to before, more than before, much more than before. The remaining items concerned inappropriate elimination, destructiveness, manifestations of reactivity, and anger. The symptoms analyzed in this survey were chosen because they are indicators of a possible anxious state [20].

### 2.5. Owner Perception of Distress and Emotional Discomfort in the Dogs during the Quarantine

Owners were provided four items on the perception of their dog’s distress due to lockdown restrictions. The four items were distributed in different parts of the survey. The choice and formulation of the items were intended to check the coherence and accuracy of the participants’ responses in compiling the survey.

### 2.6. Data Analysis

#### 2.6.1. Statistical Analysis

Statistical analyses were performed using the R language v.3.6.3 (Boston, MA, USA) and the RStudio environment v.1.2.5033 (Boston, MA, USA) [21], employing a statistical significance at *p* = 0.05. The libraries used in the current study were “psych” [22] and “GPArotation” [23]. Descriptive statistics were computed to present sample characteristics. Item distribution was evaluated in order to verify a normal multi-distribution of the scale, using the “MVN” library [24].

#### 2.6.2. Internal Consistency

For reliability estimation, the internal consistency approach (Cronbach alpha) was used. According to scientific literature, a value over 0.70 can be considered satisfactory [25].

#### 2.6.3. Construct Validity

An EFA was conducted with oblique rotation (oblimin). Because Kaiser has been criticized and considered problematic [26,27], Horn’s Parallel Analysis was performed in order to determine the number of the factors to be retained. In Horn’s Parallel Analysis [28] the observed eigenvalues extracted from the correlation matrix are analyzed with those obtained from uncorrelated normal variables. The method uses the Monte Carlo simulation process, since ‘expected’ eigenvalues are obtained by simulating normal random samples that parallel the observed data in terms of sample size and the number of variables [29]. A factor-loading coefficient of 0.30 or higher was chosen [30]. The fit indices’ results were evaluated following the conventional criteria [31]: RMSEA value below 0.06 and SRMR value below 0.08. The sample size estimation for factorial analysis was set at a minimum of 100 patients, based on the number of items composing the scale [32].

#### 2.6.4. Differences between the Means

The dependent measures, namely, the factor revealed by the construct validity analysis, were analyzed employing a paired sample *t* test, considering the two times of the survey: “T1“, namely “before the quarantine”, and “T2”, namely “during the quarantine”.

Data were analyzed using IBM SPSS version 25 (IBM Corp, 2017, Armonk, NY, USA). The level of significance was set at *p* = 0.05. Meanwhile, p levels between 0.05 and 0.10 were considered as marginally significant.

## 3. Results

Our sample was represented by 2028 participants, of whom 1258 provided complete answers to the survey, while 668 answered partially the items of the questionnaire, and 102 were disqualified, while 4858 visited the survey site. Missing values were processed with the listwise approach. Our sample was made up by 1338 (81.39%) women and 306 (18.61%) men (Figure 1A). The demographic data are reported in Figure 1.

The survey [33] consisted of a first part in which general information about the owner and the context of life were reported, followed by a part with general information about the dog. In addition, the Monash scale was used to assess the owner-dog relationship before and during quarantine. Finally, the last part concerned the frequency or intensity of the physiological and behavioral manifestations of dogs before the quarantine compared with the actual period (i.e., during quarantine). Socio-demographic data were used as statistical control for the analysis (i.e., covariates or between-group factors).

### 3.1. General and Demographic Information on Dog Owners

Figure 1 reports data about sex (see above), age (B), job (C), and level of education (D) of the interviewed sample. The participants were aged as follow: 406 were 30-40 years old (24.70%); 359 were 20-30 years old (21.84%); 358 were 40-50 (21.78%); 352 were 50-60 (21.41%); 151 were 60-70 (9.18%); and 18 were over 70 (1.09%). About the job, 541 (32.91%) were employed, 209 were freelance (12.71), 193 were students (11.74%), 191 were workers in the dog field (11.62%), 140 were unemployed (8.52%), and 370 were other (22.51%). The educational level of the sample was: college, 787 (47.87); high school, 770 (46.84%); and primary school, 87 (5.29%).

The composition of the family unit was made up as follows: 33.52% of two people; 22.57% of three people; 21.41% of four people; 16.85% of one person; 4.56% of five people; and 1.09% of six or more people.

A very high percentage (97.98%) of the participants stated that in their own family there was no positive COVID-19 case.

In relation to housing, the interviewed sample said 43.42% lived in a house between 66 and 110 sqm, 24.36% in houses between 110 and 140 sqm, 16.70% in houses over 150 sqm, while the 15.56% in houses up to 65 sqm; 85.82% had balconies or terraces associated to their homes, while 69.57% declared to have green areas available (in common or exclusive use).

### 3.2. General Information of the Dogs

Participants stated that 65.03% had one dog, 22.62% had two dogs, 7.12% had three dogs, and, finally, 2.90% had four dogs; 66.71% of the interviewed sample declared having no other animal.

A majority of the participants (60.31%) stated that their dog lives both inside and outside their house, 35.16% said it lives only inside, and 4.58% said it lives only outside.

The sex of the dogs of the present study is reported in Figure 2, where the percentage of intact or neutered dogs is considered, too. Males were 751 (48.57%), of which 498 (66.31%) were intact, while 253 (33.69%) were neutered. Females were 795 (51.43%), of which 214 (26.92%) were intact, while 581 (73.08%) were neutered. Furthermore, the dogs weighed up to 10 kg for 26.20%, from 11 to 20 kg for 30.14%, and over 21 kg for 43.66% of participants.

A part of the owners (23.90%) declared that they took the dog from a kennel or shelter, 20.97% adopted it from another person, 14.11% collected it from the street, 13.98% purchased from a private person, 17.84% purchased from a dog farm, 2.60% bought it in a store, and 6.60% reported other means.

In the present survey, 62.69% of owners stated that the dog’s age at the time of adopting the dog was between 2 and 3 months old, 15.14% said it was between 4 and 6 months old, 6.97% said it was between 6 and 12 months old, 9.04% said it was between 1 and 3 years old, 3.78% said it was between 3 and 5 years old, and 2.48% said it was over five years old.

No problems in dog management were found by owners after adoption in 80.09%. Of the rest (19.01%), 49.66% of owners found the appearance of an anxious emotional attitude in the dog, 48.99% reported fear-related problems, 33.89% reported destructiveness problems, 22.82% reported aggression, 19.80% reported inappropriate elimination, and 9.06% reported avoidance. Within the dogs with the described issues (19.01%), 60.05% had multiple issues (e.g., avoidance and destructiveness).The owners were asked if the dog had attended a dog trainer center: 61.12% of owners answered negatively; while for the rest (38.88%), 52.23% answered that the dog was followed by a dog educator, 19.18% answered that the dog was followed by a dog instructor, 15.58% answered that the dog was followed by dog trainers, and 13.01% answered that the dog was followed by veterinarians with experience in behavior. (In Italy there is a difference between dog educator, dog trainer, dog instructor, and veterinary expert in behavior. A dog educator takes care of the training during the growth and the management of adult dogs, without behavioral problems, like a schoolteacher. A dog trainer develops particular skills and performances in the dog, through certain disciplines, like a coach. A dog instructor takes care of re-educating the dog, assisting in its rehabilitation when a behavioral pathology has been diagnosed by a behavioral veterinarian and works together with a veterinary expert in behavior. A veterinary expert in behavior graduated with a degree in veterinary medicine and has followed a path of specialization, theoretical and practical, thanks to which he/she is able to diagnose behavioral pathologies. He/she is, therefore, a doctor with ethological, clinical, and zoopsychiatric skills, who can then make a diagnosis, identify a given behavioral pathology, and establish a prognosis, identifying objectives and useful times for rehabilitation therapy). The majority of owners (88.08%) stated that their dog did not suffer from chronic pain, and 93.74% declared their dog had never been diagnosed with a behavioral disorder.

### 3.3. Monash Scale

#### 3.3.1. Internal Reliability

The reliability of the adapted version of the “Owner-Dog Interaction” and “Perceived Costs” subscales was evaluated using Cronbach’s alpha coefficient, and all the values were greater than 0.90.

#### 3.3.2. Construct Validity

To investigate the psychometric validity of the tool, we conducted an Exploratory Factorial Analysis EFA with a principal axis factoring method (PAF) and “oblimin” rotation on the 10 items. EfA is generally used to discover the factor structure of a measure and to examine its internal reliability.

Prior to conducting an EFA with PAF, Bartlett’s test of sphericity and the Kaiser-Meyer-Olkin measure of sampling adequacy (KMO) were evaluated. The correlation matrix showed that most of the items had a correlation greater than 0.35. Bartlett’s test of sphericity was significant (*p* < 0.001), and the KMO value was >0.6 (KMO = 0.70), indicating that factor analysis was appropriate for the data [34].

Parallel analysis was undertaken using the Horn’s procedure. Using 5000 parallel data sets, α = 0.01, parallel analysis indicated a two-factor solution. Subsequently, principal axis factoring resulted in two factors accounting for 43.0% of total variance. One item (“Quanto spesso dai da mangiare al tuo cane premi alimentari?” “How often do you feed your dog food prizes?“) resulted with a factor loading ≤0.3 and was deleted from the analysis. Fit indices for the EFA were considered acceptable (RMSEA: 0.06, TLI: 0.94, SRMR: 0.03), which indicated a consistent factor structure.

Table 1 depicts the relationship of each item of the adapted MDORS to the underlying factor (factor loadings). Factor loading helps the understanding and interpretation of the set of items that cluster on the same factor.

Considering the items that clustered on the same factors, we named the factor 1 (F1) as “Owner-dog interaction” and factor 2 (F2) as “Perceived cost”. Considering the original structure of the MDORS scale, the current analysis showed a similar structure (“Owner-dog interaction”, “Perceived cost”).

#### 3.3.3. Perceived Difference between before and during the Quarantine

After checking for the normality of the data (skewness and kurtosis ranges between −0.9 to 1.1), results of the paired sample *t*-test on the “owner-dog interaction” factor showed a significant, but little, difference between before and during the quarantine period (t = 6.3505; df = 1272; *p* value *<* 0.001; C.I. 95% = 0.15-0.30; mean of the differences 0.23; effect size = 0.18). Considering the “perceived cost” factor, the differences between before and during the quarantine showed a significant and evident difference (t = 28.866; df = 1270; *p* value < 0.001 C.I. 95% = 1.93-2.2; mean of the differences *=* 2.07; effect size = 0.81). See Figure 3A,B for graphical information of the descriptive statistics. Summarizing, considering the interaction, even if statistically significant, the very small effect size showed that the interaction did not change a lot between before and during the quarantine, while the owners perceived their dog during quarantine as less challenging than before (i.e., perceived cost), with a significant and consistent difference. Furthermore, we statistically evaluated, through a series of ANOVAs, the influence of sociodemographic characteristics considering the timing of the survey as a within factor (pre-post) and the different individual characteristics as a between factor. The only significant negative interaction effect emerged considering “having green area in the owner’s house” on the perceived costs (df = 3; F = 2.83; *p* = 0.03), where having green area reduced even more the owners’ perception of the dogs’ costs.

### 3.4. Information on Dog’s Lifestyle during the Quarantine

Owners were asked questions about any changes in the dog’s lifestyle during the quarantine period, with Yes or No responses. The results are shown in Figure 4.

During the quarantine, the family unit was unchanged for 88.19%, decreased for 5.94%, and increased for 5.87%. No dog (100%) was temporarily placed in kennels or boarding houses.

With respect to intraspecific sociability, to the question, “If on a walk you meet another dog that interacted with your dog before quarantine, will you allow your dog to get close?”, the sample replied: less than before for 37.17%, much less than before 13.70%, never 13.55%, more than before 13.96%, much more than before 3.62%.

Upon returning from walks, 27.68% of owners said the dog was cleaned differently than before. Participants were asked, “How is usually the dog cleaned?”. The answers were as follows: 48.84% cleaned the dog with dog wipes, 18.91% with a damp cloth, 11.30% with soap and water, 3.26% with disinfectant, and 17.78% other. The owners were asked if access to sofas and beds had been restricted to dogs for fear of spreading the virus, and the 97.68% answered that there had been no limitation. A question was asked for information regarding any cessation of sporting or playing activities with respect to prior to quarantine: 45.87% of the sample stated that they had to give up such activities.

### 3.5. Information on Clinical and Behavioral Symptoms in the Dog during the Quarantine

Items about clinical and behavioral symptoms were evaluated. Of these, 11 items provided as an answer: “much more than before, more than before, as before, less than before, much less than before”. Data are reported in Figure 5. The remaining items had a positive or negative answer.

Regarding behavioral changes related to situations of emotional distress, 95% of owners stated that the dog did not start inappropriate elimination at home and 97.25% that the dog did not start destroying things at home.

Data relating to the question, “Did the dog start having difficulty being alone?”, the majority of the sample stated that no difficulty was detected in their dog, while 17.39% stated that the dog began to have difficulties being alone. The dogs of the latter group exhibited the following behaviors if left alone: 38.56% whining, 33.47% barking, 6.36% destruction, 1.69% sialorrhea, and 19.92% reported other behaviors in the dog.

The owner was asked if the dog was more reactive than before. The sample that responded in the affirmative (19.35%) stated that the dog exhibited greater reactivity toward unknown people (20.60%), toward family members (32.58%), toward other dogs outdoors (29.21%), and other people (11.24%). Additionally, owners were asked if and how the dog manifested anger. They answered as follows: The dog does not manifest anger (57.61%); barks (27.32%); and growls (9.13%).

### 3.6. Owner Perception of Distress and Emotional Discomfort in the Dogs during the Quarantine

A part of the survey was organized to collect data to understand how the owners perceived any discomfort of their dogs during the quarantine, and what were the changes of the dogs’ lifestyle during quarantine compared to before.

When asked, “How do you think your dog’s quality of life was during this quarantine period?”, the sample responded in relation to a scale from 1 to 10, where 1 corresponds to “low quality” and 10 to “high quality” of life (Figure 5).

Owners reported, for 80.36%, that their dog did not experience the changes due to quarantine with discomfort, while the rest (19.64%) said the opposite.

To the question, “In your opinion, how much does the current quarantine condition affect the dog’s discomfort?”, 41.10% answered “little”, 25.22% answered “nothing”, 23.12% answered “a fair amount”, 6.88% answered “much”, and 3.77% answered “I don’t know”.

To the question, “How much does this quarantine situation worry you about your dog?”, the sample replied as follows: 45.65% “little”, 28.62% “nothing”, 21.30% “a fair amount”, and 4.42% “much”.

## 4. Discussion

In a moment of fragility and uncertainty caused by the SARS-CoV-2 pandemic, the present survey aimed to offer a contribution to the study of the human-dog relationship by evaluating the discomfort experienced by the dog and how it was perceived by the owner. In fact, the survey targets the assessment of interaction and perceived costs in the context of the relationship (Monash scale) and the intensity of behavioral symptoms manifested in dogs.

Our survey was run online on 8 May 2020 and the data collection took place from 8 to 18 May 2020. The choice of this period was motivated by the fact that we found it more appropriate to wait until the final phase of most restrictive impositions. The reasons for the choice were the following:(1)Fear that the reading of the dog’s discomfort may be compromised by a state of psychological discomfort of the owner and that this discomfort may affect an objective reading of the dog’s discomfort. In fact, it has been shown that stress and poor welfare of owners negatively affect the well-being of their pets [35].(2)To allow owners to report the increased or decreased frequency of symptoms during the whole lockdown compared to the previous period. In fact, we believe that having a survey carried out in the first weeks of the lockdown could have given unreliable answers. In fact, a modified intensity of the symptoms could be attributable to the temporary adaptation phase of the dog due to the change of routine that the lockdown imposed [20].(3)In order to obtain complete feedback from the owner regarding the entire period of maximum restriction in relation to the Monash scale: interaction vs. perceived costs.

### 4.1. Dog Owners

Our sample, for the most part, was represented by women (Figure 1A). This is not a surprise; in fact, it is in line with what has been reported in several studies [36,37]. In a study on communication between veterinarians and clients, Shaw et al. [38] showed women are more involved in pet management and, as reported by Smith [39], women are more likely than men to participate in online surveys. In support of this hypothesis and as found by Christley and collaborators [40], women and younger owners are more likely to spend more time at home than men and older owners [38].

About the job professions of sample, it is interesting that 11.62% of the sample worked in the dog and veterinary field. These participants represent a population that might be more aware of a human-dog bond and potential measures needed during lockdown, for example, adopting measures to ensure a greater degree of well-being and avoiding particularly stressful situations for the dog, from a physical, nutritional, and behavioral point of view. In agreement with Bloom [41], experienced people were better at identifying behaviors of dog (*Canis familiaris*) facial expressions from photographs. Several studies suggest that children and adults do not reliably understand the body signals of dogs [41,42] and that people often mistake angry dog facial displays for happy ones [43]. Indeed, not all emotions may be equally easy to recognize. Overall, people are generally more successful at recognizing positive dog emotions, like happiness [41,44,45], while often confusing negative emotions, like fear [41,44,45,46]. With respect to the level of education, our sample placed in a medium-high range, in which graduates represented 47.87% (Figure 1D). Literature shows a correlation between socio-cultural level and greater resilience in facing adversity due to greater cognitive tools [47]. As the sampling of our study was voluntary, the possibility of selection bias could not be eliminated, considering also the use of a web-based survey. [48]

A major portion of owners was able to use green areas for private use and for their dog, even during the restriction period. This is particularly important for the owners, as it allowed an easier management of the dog (play, eliminations, movement) [49] despite the restrictions of the Decree [16].

This is particularly important in relation to the prohibition moving away over 200 m from the house to take the dog for a walk, as established by the decree #IoRestoACasa [16]. This aspect drastically changed the routine of most dogs.

### 4.2. Dogs

With respect to the sex of the dogs, a higher percentage of neutered females was found compared to neutered males. The sterilization surgery in the female is more invasive, with longer recovery times, and more expensive than the same surgery in the male dog. Despite this, the owners preferred to sterilize the females rather than the males (Figure 2).

As expected, a significant percentage of owners (62.69%) adopted their dogs at the age of 2–3 months. With the increase of dog age, the percentage was drastically and progressively reduced, up to 2.48% for dogs adopted at the age of more than 5 years. It is common belief that the younger the dog is at the time of adoption, the easier it is to structure a functional attachment bond [17]. However, this does not exclude the ability to create multiple bonds of attachment during the dog’s life that characterize the two-way human-dog relationship [50]. Furthermore, a gender-dependent difference was highlighted with respect to age at the time of adoption. In fact, women are more likely to adopt older dogs than men, proving the strong ability and motivation of women for caring [50]. The drift of caregiving can explain the higher percentage of women involved in dysfunctional relationships with animals, such as animal hoarding [51].

The relationship and the size of the attachment influence the degree of social reference between dog and owner. This relationship allows dogs to interact safely in the presence of the owner and to show less distress in response to threatening events [52]. The owner is, to all intents and purposes, the figure of reference and care for the dog [53].

The percentage of dogs aged 0–2 years related to this questionnaire was 19.68%. This age group, which includes puppies and adolescents, is the most at risk, as the limitations imposed by the lockdown could have negative repercussions on normal behavioral development. The reasons are basically two: (1) the environmental intra- and interspecific and social hypostimulation that the puppies have undergone and (2) the constant presence of people in the house, which did not favor the correct processing of the detachment. All of this could result in poor emotional management of the dog when left alone [38]. In fact, the concept of ontogenetic development refers to the modifications over time of the perceptual and behavioral properties of the subject [54]. During the progressive physiological and behavioral development, the individual is enriched with sensory, emotional, cognitive, and social skills that will affect the future life.

The low percentage of dogs that live exclusively outside the house (4.53%) suggests the high relational levels and how much the sample is attentive to the ethological needs of his/her dog. In fact, the dog, being a highly social animal, needs to feel part of a group, with whom to share its life. Sharing spaces in the context of a human-dog relationship is essential, as they are both social species. Several studies have been published on dog social cognition, focusing on behavioral synchronization in the dog-human dyad [55,56,57]. The dog that daily shares life with human beings acquires cognitive and social skills in relation to the dimensions of the relationship. The high rates of attention of owners are in line with the high affective relational spheres reported in the Monash scale. The high attention paid to the protection of behavioral development is also evident from the percentage (38.88%) of dogs that attended dog centers and were followed by an educator/dog trainer. The smaller percentage, of 13.01%, of dogs that were followed by a veterinarian with experience in behavior is also reflected in the low percentage of owners who report that their dog has been diagnosed with a behavioral problem (6.26%).

### 4.3. Dog’s Lifestyle during Quarantine

During the quarantine, the owners (61.45%) declared that there had been changes in their dog’s life. In particular, the types of changes, during the lockdown and compared to before, referred to the usual hours, the type of food, and being led on a walk by people other than the usual ones. It should be noted that no dog (0%) was entrusted to others or taken to kennels or boarding houses.

With respect to intraspecific sociability, we asked, “If on a walk you meet another dog with whom it interacted before quarantine, will you allow it to get close?”, and 64.42% of the sample (given by the sum of the percentages relating to the answers “never”, “less than before”, “much less than before”) decreased the intraspecific interactions while walking. This is attributable both to the restrictions imposed by the lockdown and to the fear of possible contagion of the disease through other dogs [1]. The lack of knowledge of the interspecific transmission dynamics of the virus, as was to be expected, led 27.68% of the sample to clean their dog in a different way than before after returning from walks. The uncertainty with respect to the role of their pets as a vehicle of contagion led owners to incorrectly use aggressive disinfectant products. In fact, many dogs, with dermatitis or burns caused by chemical disinfectants, visited the veterinary emergency room. Such circumstances led the FNOVI (National Federation of Italian Veterinarian Orders) to formulate guidelines to dissuade people from using sodium hypochlorite and other disinfectants. The state of increased attention from a hygienic point of view was not supported, however, by the answer to the question, “Compared to before, have you limited your dog’s access to beds and sofas for fear that it may spread the virus?”; only 2.32% answered in the affirmative. These findings are confirmed by what was reported by Pawling et al. [58] with respect to the fact that the tactile element of social interactions plays a fundamental role in buffering physiological and psychological stress in humans and other gregarious species. The findings are reflected in the assessment of affectivity compared to the perceived costs of the Monash scale. Additionally, the role of the dog as social support is confirmed by Bowen et al. [36], suggesting that, during the current epidemic, the relationship between people and their pets has helped offset the drastic reduction in their social and physical interactions with people.

Forced renunciation of dog sports and play activities was reported by 45.87% of the sample. This partially may explain the increase in symptoms related to anxious manifestations. In the context of the interspecific relationship, the game seems to improve the social cohesion between humans and dogs, increasing their familiarity and reducing competitive interactions. In fact, playing between humans and dogs is important to strengthen their bond. Dogs like to play with humans and prefer to play with a human than on their own when there is a toy around. Dogs that do not get enough play opportunities when they are young may show inappropriate behavior in adult play with other dogs or humans. If it is misinterpreted by the owner as actual aggression and the dog is given fewer play opportunities as a result, this may lead to reduced welfare [59]. Furthermore, a UK study by Christley and colleagues in 2021 [40] reported that during the lockdown their dogs received more frequent playing sessions and received more toys. The increase in play and training opportunities is encouraging, as it has been noted that this enrichment provides dogs with important mental, social, and behavioral stimuli [35,60]. In Italy, during the lockdown, it was not possible to attend dog centers for sporting activities and for behavioral recovery, to go to the veterinarian for routine checks, or to provide for the usual cleaning of the dog. However, our data suggest that 27.9% of our participants reported a higher number of play requests by their dogs (Figure 5), while the interaction measured by the MDORS showed a very little decrease of owner-dog interaction between before and during the quarantine. These restrictions may have negatively affected both the dog, in terms of caregiving, and the owner. [61].

### 4.4. Behavioral Symptoms

With the term “behavioral symptoms” we intended to summarize behavioral manifestations reported by the owners in the survey for evaluation of dog well-being.

The survey had the aim to understand how the owners perceived any discomfort of their dog during the quarantine. The introduction of a section of items referred to the intensity of the behavioral symptoms as essential for obtaining objective data on the dog’s well-being during the quarantine compared to the period before the quarantine. Symptoms have an objective value in the evaluation of behavioral manifestations, compared to the perception of the well-being of dogs reported by owners. This perception may, in fact, be influenced by the emotional state of the owners [36].

There were no substantial differences in the variations in appetite and thirst during the lockdown period with respect to before.

Our sample reported that the dog yawned more than before (12.83%). Yawning is a widespread phenomenon in mammals and birds. Several physiological and social hypotheses have been formulated about the functions of spontaneous yawning in non-human animals. Indeed, yawning could act as a homeostatic brain-restoring and -cooling mechanism and a signal of anxiety and sleepiness and plays a role in the social communication system as well [62]. Therefore, in the context of the relationship, the yawn also has another value: It seems to mix with the emotional contagion, which is defined as “the tendency to automatically imitate and synchronize expressions, vocalizations, postures and movements with those of another person and, consequently, to emotionally converge” [63]. The sample reported that the dog whined more than before (14.49%). This increase can be linked to an anxious state. In fact, the restrictions imposed by the lockdown led to a change in the dog’s routine that can manifest anxiety in anticipation of an event whose failure to fulfill could lead to frustration and an increase in the search for attention [64]. Additionally, in this survey we reported a considerable increase (32.83%) of requests for interaction with cohabitants.

The sample (17.03%) reported that the dog slept more than before. This increase may be related to a reduction in physical activity imposed by the restrictions. Although research on animal boredom is sparse, boredom could be expressed in a multitude of different ways. These may include increased daytime sleepiness, sleep disturbances, and restlessness, as well as signs of frustration or sensation seeking, seeking attention, and showing increased exploratory or destructive behaviors [40]. These anxious manifestations are confirmed by the increase in hair licking (17.58%) and movement at home (17.48%). The percentages detected in our survey in relation to these symptoms were, in fact, superimposable.

A part of the sample (17.39%) reported that their dog began to show difficulty being alone. These data are in line with those reported by Holland et al. [37] about the observation of new undesirable behaviors in their dogs during the lockdown, including barking and vocalizing together with a “clingy” attitude when briefly left alone. During the lockdown the dogs had less opportunity to spend quiet time away from people. This could negatively affect well-being and increase separation anxiety when left alone. Another part of the sample (19.35%) reported that their dog was more reactive than before, especially toward family members. This finding is important and could be related to a lack of satisfaction in various areas: due to lack of physical exercise, to lack of socialization with other dogs, to forced co-habitation for longer than before, and to lack of quiet time [36,37,49].

### 4.5. Owner Perception of Distress and Emotional Discomfort in the Dogs during the Quarantine

With respect to the owners’ perception of their dogs’ discomfort, the sample was asked several non-sequential questions: (1) “How do you think your dog’s quality of life was in this quarantine period?”, (2) “Has the dog experienced or is experiencing with discomfort these changes due to quarantine?”, (3) “In your opinion, how much does the current quarantine condition affect the dog’s discomfort?”, and (4) “How much does this quarantine situation make you worried about your dog?”. With respect to the question on the quality of life (Figure 6), the highest percentages were found for values between 7 and 10, therefore, in a very high evaluation range. These data are in agreement with those reported by Esam et al. [65] in New Zealand, where the owners did not express concern about any inconvenience. The only concern reported was with the management of separation anxiety once the normal pre-lockdown routine was resumed. Additionally, in this study, the sample revealed high attention to the quality of life of their animals.

In the present study, the perception of a dog’s high quality of life was confirmed by the percentages found in the other three questions. The percentages of the answers to these questions were superimposable, which confirmed the consistency of the answers of the interviewed sample. These items had a control function to verify the reliability of the sample. From the analysis of these responses, a certain optimism of owners emerged with respect to the risk that the limitations imposed by the lockdown could have a negative impact on the welfare of their dogs.

The percentage of owners who, in the item about their dog’s quality of life (Figure 6), responded in the lower evaluation range was in line with the increase in the percentages of behavioral symptoms related to anxiety (Figure 5).

## 5. Conclusions

The data that emerged from this survey help us to better understand the complexity of the bond that characterizes the human-dog relationship. The dog, in fact, makes use of a reference figure to overcome complex situations and anguish. It is also true that humans, through a process of emotional osmosis, derive pleasure from the presence of the pet and the affective areas that distinguish the relationship with our pets.

This picture is in line with the One Welfare approach, which implies the existence of a two-way connection between the well-being and health of humans and non-human animals [66].

## 6. Limitations

Using a representative sample, we were able to make a general characterization of the well-being of the dogs, trough the evaluation of the symptoms by the owners, and an evaluation of the interaction between owners and dogs during the different phases of the pandemic. However, one main limitation is the recall bias, since we were unable to recruit the sample before the pandemic; therefore, we had to recollect the same measures asking also about events or habits related, on average, to two months before. These were compromises that we felt were worthwhile, given the opportunity to collect data during such an unusual event, but they limit the value of the data. However, literature has demonstrated that a short period seems to reduce the risk of recall bias [67]. Future studies should address these limitations, using longitudinal research designs. During the lockdown we also did not explore the relationship between individual circumstances and the social support obtained. For example, how the type and level of social support from the dog related to the quality of a person’s wider social support network, the stresses the person experienced, their physical and mental health, and how technological solutions (such as video calling and social media) had mitigated the non-physical aspects of social isolation. These are areas that require further study.

## Figures and Tables

**Figure 1 animals-11-02335-f001:**
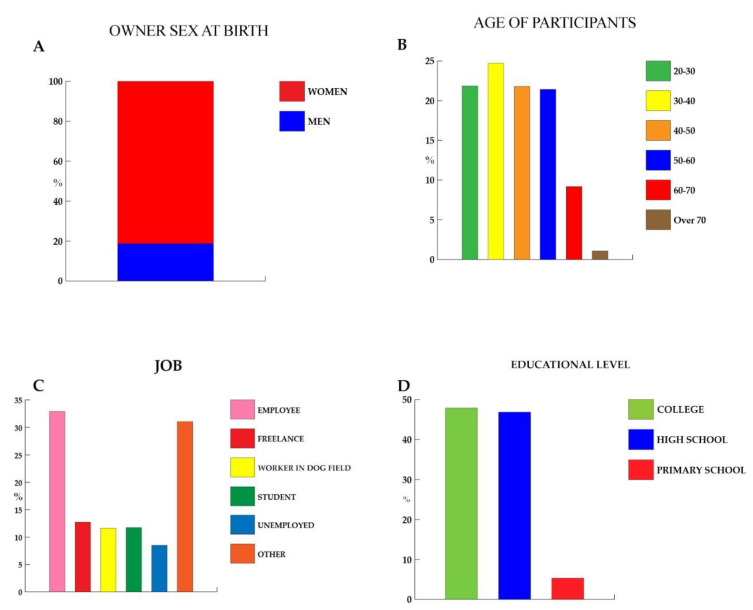
Information on dog owners. (**A**) Participants’ sex; (**B**) participants’ age; (**C**) participants’ job; (**D**) participants’ degree. Data are expressed in percentage.

**Figure 2 animals-11-02335-f002:**
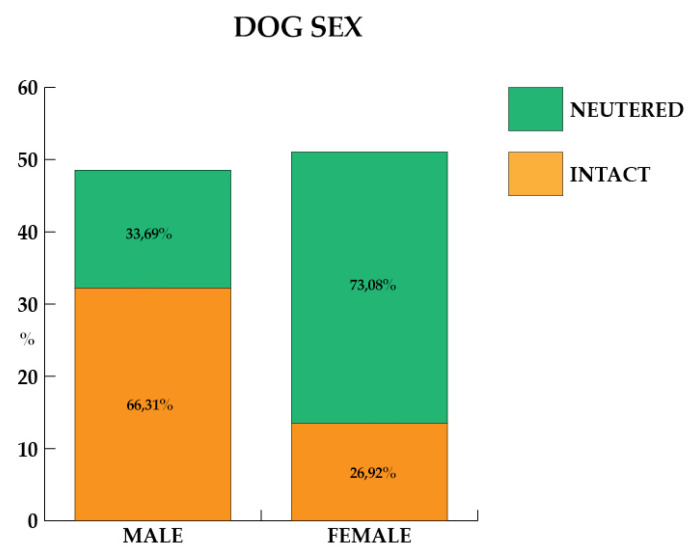
Percentage of dogs’ sex, distinct in intact and neutered dogs.

**Figure 3 animals-11-02335-f003:**
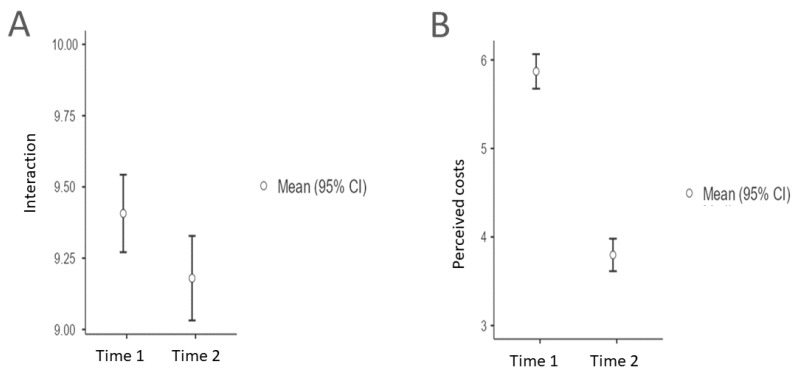
(**A**) Differences in the “Owner-dog interaction and love” factor between the two time of the study. (**B**) Differences in the “Perceived cost” factor between the two times of the study. Time 1 is before the quarantine period; Time 2 is during the quarantine period.

**Figure 4 animals-11-02335-f004:**
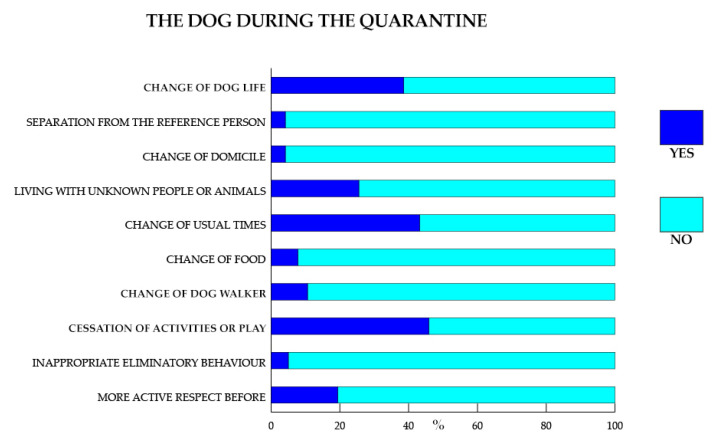
Dog’s lifestyle during the quarantine. Data are expressed in percentage.

**Figure 5 animals-11-02335-f005:**
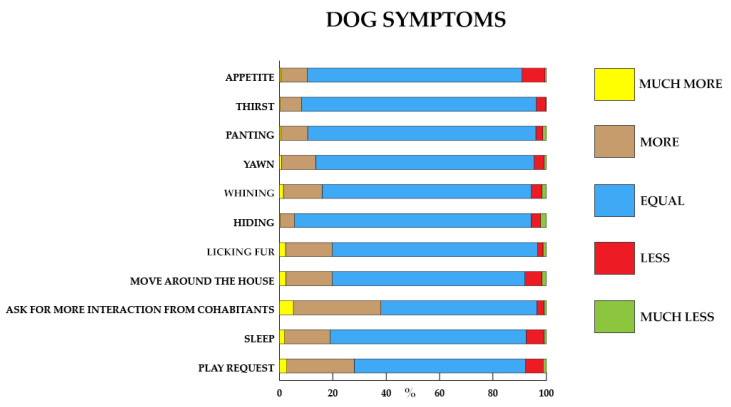
Clinical and behavioral symptoms manifested by the dog and reported by the owner. The answers included a comparison between the period before quarantine and the period of quarantine. The answers were “much more than before”, “more than before”, “equal to before”, “less than before”, and “much less than before”. Data are expressed in percentage.

**Figure 6 animals-11-02335-f006:**
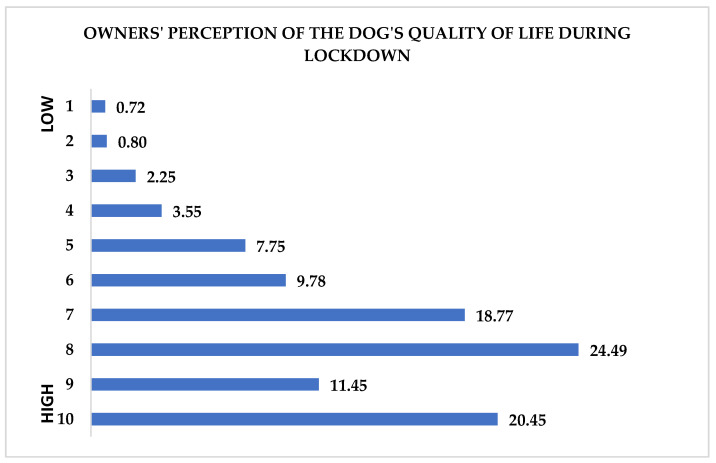
How the owners perceived their dogs’ quality of life. The sample responded in relation to a scale from 1 to 10, where 1 corresponded to “low quality” and 10 to “high quality” of life. Data are expressed in percentage.

**Table 1 animals-11-02335-t001:** Factor loading for the two subscales of the MDORS used.

Items	F1	F2
How often do you hug your dog? *(Quanto spesso abbracci il tuo cane?)*	0.871	
How often do you kiss your dog? *(Quanto spesso baci il tuo cane?)*	0.748	
How often do you play with your dog? *(Quanto spesso giochi con il tuo cane?)*	0.340	
It bothers me that my dog stops me doing things I enjoyed doing before I owned it. *(Mi infastidisce che il mio cane non mi fa svolgere alcune attività che mi piaceva fare prima di possederlo)*		0.709
It is annoying that I sometimes have to change my plans because of my dog. *(E’ scocciante dover cambiare i miei piani a causa del mio cane)*		0.707
There are major aspects of owning a dog I don’t like. *(Ci sono degli aspetti rilevanti del possedere un cane che non mi piacciono)*		0.745
How often do you feel that looking after your dog is a chore? *(Quanto spesso senti che prendersi cura del tuo cane è un impegno?)*		0.476
My dog costs too much money. *(La gestione del mio cane costa troppo)*		0.381
How often does your dog stop you doing things you want to? *(Quanto spesso il tuo cane ha interrotto o impedito le attività che avresti voluto fare?)*		0.301

## Data Availability

Data set, Survey Report and Survey in English are available by clicking the following link: https://osf.io/5m6wb (accessed on 6 August 2021).

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
