# Peer review of "Human-Dog Relationship during the First COVID-19 Lockdown in Italy"

_animals, 2021, doi:10.3390/ani11082335_

Round 1

Reviewer 1 Report

Article Review (Journal): ANIMALS
Manuscript ID: animals-1253921
Title: Human-dog relationship during the first Covid-19 lockdown in Italy: more love, less efforts and perceived costs

Reviewer recommendations: Major revisions

Thank you for your very interesting research article about the Human-dog relationship during the COVID-19 pandemic lockdown. The article is of good quality and has relevant information to add to the current literature. Please see below for specific details that may further enhance the quality of the work.

SIMPLE SUMMARY

When you read the conclusions on the simple summary, it is not easy to understand the meaning of “perceived costs”. It should be rewritten to facilitate the reader comprehension.

“The survey analysis highlighted an increase in affectivity and a decrease in perceived costs in the human-dog relationship”.

ABSTRACT

Should not contain abbreviations (line 37): MDORS. 

You introduce the abbreviation "MDORS" but never define this as " Monash Dog Owner, Relationship Scale".

Abstract should follow the results and conclusion from the main text. The sentence “The results highlighted how affectivity prevailed over perceived costs despite the period of great economic difficulty”. Is not stated in the results (there is nothing in the research that states great economic difficulty) and conclusion.

INTRODUCTION

Line 69 the word “learned” is misspelled as “learnt”.

RESULTS

There is a mistake on the figure 1 description (Line 207): “Figure 1. reports data about sex (A), job (B), age (C) “ it should be “Figure 1. reports data about sex (A), age (B), job (C)  “

It would be interesting to add the total number with the % for all the results.

Line 229. The word furthermore is not necessary

Line 248. Substitute “has” to “had”…. “dog has never been diagnosed with”.

The table 1 results are very confusing and cannot be comprehended, even reading the text.

“Table 1. Factor loading the two subscales of the MDORS used” substitute “subscale” for “subscales”

Line 277 and 281, if it is statistically significant, there is no point of saying little or evident difference “factor showed a significant but little difference”

Figure 2: add total number (n)

Figure 3: Figure legends need to be auto explanatory. Include the meaning of t1 and t2, and label the Y axis.

DISCUSSION

All discussion sections need to be reviewed since the results are being stated in the discussion part.

Line 369 vs is bold “affectivity vs. perceived costs”.

Results are misplaced into the discussion section. For example, “About the job of sample, it is interesting that a part works in dog and veterinary field 378 (11.62%) (Figure 1C).”  No discussion has been added to the result stated in the discussion section.

I am curious: Did the survey looked if the owners were staying home longer times due to the lockdown? With the lockdown, dogs were left alone for shorter periods of time, which probably increased their quality of life.

CONCLUSIONS:

The authors cannot conclude “a transient increase in dog abandonment occurred due 534 to health, economic and social stress and the fear that the dog could be a viral vector”. Because there is no results on that. They can discuss it, but not conclude.

Aims: 1. investigate the human-dog relationship in a social isolation

  1. quantification of symptoms, the resilience of the dog during the lockdown

There is no conclusion added to the main manuscript, according with the two aims proposed by the authors.

Author Response

Thank you for your comments, we modified the manuscript according to them.

Reviewer 2 Report

See attached for detailed notes.  The main issue is the need for a point that travels throughout your paper from introduction to results and discussion.  It feels wandering and tangential currently.  There is not enough focus on the limitations of the conclusions of this retrospective study.

Author Response

Thank you for your valuable comments, please see the attachment.

Reviewer 3 Report

REVIEW FOR ANIMALS

Title.

Human-dog relationship during the first Covid-19 lockdown in Italy: more love, less efforts and perceived costs

General comments

A timely and interesting study that reveals some interesting aspects about the human-dog bond during COVID. I found this study very interesting, well considered and well analysed – thank you for the opportunity to review it.

The overall work is of a high quality and is well constructed and generally well written. There are occasional lapses in language and style that affect clarity and conciseness in areas; I have made some comment about this below and proofing is recommended.

I hope you take these comments in the supportive spirit in which they are intended. Suggested amends are to enhance the clarity of the work and to enhance ‘readability’ and accessibility.

Title – consider changing ‘efforts’ to effort for clarity and ‘correctness’ (although the term ‘efforts’ might be preferred by authors!) – this also applies within the work where the term’ efforts’ is used. If referring to social Exchange theory, then possibly consider use of benefits and costs to align more clearly. It is however entirely possible that I am misreading your intent here!

9 – indented space on affiliation - formatting

19 – ‘more or less long period’ – Suggest reword for clarity

34 - ‘more or less long period’ – Suggest reword for clarity

45 – intro sentence needs context that quarantine was undertaken because of the COVID-19 pandemic, otherwise the sentence lacks full contextual importance.

45 – consider replacing ‘states’ with ‘countries’

47 – suggest removing ‘until today’ and leaving rest of sentence

80 – insert ‘Italian’ before government for clarity

96 – 101 – check tenses used – mixing of past and present

104 – suggest reword to ‘available between the 8th and 18th May 2020, the time that corresponds to the beginning of phase 2

108 – replace regards with regarding

113 – heading is a statement not a heading? Consider reviewing.

123 – consider inserting ‘an’ before Italian

126-127 – fragmented and affects clarity.

141 – is this the study questionnaire or the MDORS? Clarify. Appendix 1 not seen.

145-146 – fragmented sentence

148 – insert ‘and’ between home and changes

161 – review use of ‘moments’

164-194 – data analysis appears well considered and appropriate. IMHO analyses are correct in terms of analysis and corrections.

214 – replace to live with lived

216 – replace stated to have with had

220 – fig 1 – nice figures but could be larger to enhance clarity

223 – replace ‘a’ with one (dog)

224 – replace to have with having

226-227 – reword for clarity.

(General comment -demographic info is interesting and potentially relevant for further discussion – could be presented in table form for accessibility and clarity – could also support better demonstration of key demographic data that might be interesting. Also – any info on breed/type?? This could be significant in parts)

230 – add ‘of participants’ at end of sentence

236 – replace ‘life’ with age

239-245 – what percentage of dogs had multiple issues?

244-246 – lacks clarity in terms of professional contact – what context is this referring to? Vet issues/medical issues? What is an educator dog trainer versus an instructor dog trainer? This distinction might be important with regards to Human-dog bond

248 – missing ‘a’ before behavioural disorder

284-286 – again consider use of ‘efforts’ as I am not sure it fully aids clarity of intent here (do you mean benefits?)

286 – insert points after time for clarity

298 – cleaned

300 – replace the with and at end of sentence

302 – renunciation is fine but could be amended to cessation or similar for clarity (also applies in later text and figure 4)

316 – ‘weeping’ – clarify meaning here

Figure 5 – lovely figure but rather small!

357 – tenses – replace is with was

378 – ‘a part’ – reword for clarity – also is the level significant? Does it represent a bias in the sample population? Does it suggest a level of self-interest and self-selection? Could this proportion of the participants represent a population that might be more aware of human-dog bond and potential mitigation measures needed during lockdown (for example I had mitigation measure in place for my own dogs within 24 hours of lockdown…… physically, nutritionally, behaviourally etc)

380-382 – would benefit form further justification of this statement – also fragments.

386 – is there a missing ‘not’ when referring to walk your dog? Clarify please.

389-391 – is this relevant? Could be better linked to demographic discussion data and also my earlier point about breed/type.

392 – was this really significant OR is the word higher a better choice

406 – incorrect reference number? 501?? Should this be 51?

416 – wording – ‘correct construction’?

417 – wording – ‘of the posting then’

421 – wording – ‘trace the future’ – clarify

422-427 – I think this merits some further discussion about dog status in family and living situation. Possibly will also link to purpose of dog and bond that human has with dog…..

434 – replace have with had (review tenses throughout for consistency)

434-439 – again this merits further consideration as could simply be a consequence of lockdown limitations rather than affectivity….. in my country, we were limited in this potential so dogs could not really be walked by others/board/go to day-care etc….

463-464 – consider adding more detail here about the potential positive impacts of playing with/interacting with dogs during lockdown – lots already published! Also consider the negatives of canine caregiving – again published too – worry over vet access etc.

465 – is symptoms the best term for clarity?

464-471 – this section is a little confused and lacks clarity of intent/content.

472-473 – confused and contradictory.

482 – do you have data on how much owners were aware of behaviour like yawning pre lockdown???

483 – ‘peeps’ – consider rewording

500-502 – reword for clarity.

503-504 – discuss why this is important (as it could well be!)

518 – typo – bof should be of

533-544 - this seems not to fully conclude from the discussion and study IMHO! 528-532 reads as a better conclusion!

560 onwards – some typos and formatting errors.  I have also not checked use and correct inclusion of references

Author Response

(The authors gave the same response as above.)

Reviewer 4 Report

Thank you for an interesting paper that uses a modification of the Monash Dog Owner Relationship Scale to evaluate dog-human social relationships during a period of lockdown in Italy.

However, I have some comments about the methodology and the interpretation of the results.

  • You used only part of the MDORS, which you did not really evaluate in your discussion. I was unable to see the full survey used (except by visiting the actual survey online, in Italian). It would be useful to see the full survey in English.
  • A large amount of demographic data was collected but it did not seem to be utilised (e.g. family make-up, building size). Perhaps you could comment on this in your results.
  • The use of two time points was a key part of the analysis and led to some important conclusions. However, it was not clear how these two time points were translated in the survey – were ‘before’ and ‘after’ lockdown used as the two time points? And if so, how reliable are the resulting data if merely reliant on the respondent’s memory? (Again, I need to see the translation of the survey to judge this properly).
  • You have used some high-powered statistical methods for testing the validity of the instruments used, but many of the results seem to rely on dichotomous yes/no  answers and Likert-type scales. Were the demographic data completely ignored in the analysis?
  • I have put comments about individual sections on the manuscript.

Author Response

(The authors gave the same response as above.)

Round 2

Reviewer 2 Report

The authors made many changes that helped to clarify the point and goals of the study.  I think that the authors should still consider whether they wish to discuss a point (e.g. the cleaning of the dog) and if so make sure that they throughly address that issue (their response to me was good but it is not clear in the paper itself).  Consider that for all the many variables that you are looking at.

Watch for grammar issues throughout.  Continue to look for sentences starting with numbers and commas instead of periods in numbers.

Line 42-44: Unclear

Line 102: evaluated

Line 112-114:  You might want to clarify that the survey was just after the strictest lockdown concluded...

Line 199-200: Rework

Author Response

We thank the Reviewer for his valuable comments and suggestions.

Q1. The authors made many changes that helped to clarify the point and goals of the study.  I think that the authors should still consider whether they wish to discuss a point (e.g. the cleaning of the dog) and if so make sure that they throughly address that issue (their response to me was good but it is not clear in the paper itself). Consider that for all the many variables that you are looking at.

A1. Thank you, we have changed the sentence, accordingly to your suggestion (see L. 487-493).

Q2. Watch for grammar issues throughout.  Continue to look for sentences starting with numbers and commas instead of periods in numbers.

A2. Yes, you are right, there were some inaccuracies that we corrected.

Q3. Line 42-44: Unclear

A3. You are right. We reworded accordingly.

Q4. Line 102: evaluated

A4. Thanks, we changed accordingly.

Q5. Line 112-114:  You might want to clarify that the survey was just after the strictest lockdown concluded...

A5. Thank you. We reworded in a cleared way (see L. 107-109).

Q6. Line 199-200: Rework

A6. Yes, we reworked the sentence accordingly (see L. 191-195).

Reviewer 4 Report

Thank you for your detailed responses to reviewers' comments. I think the revised version of the paper now reads much better.

Author Response

Q1. Thank you for your detailed responses to reviewers' comments. I think the revised version of the paper now reads much better.

A1. We appreciated your feedback, thank you for helping us improve the quality of the manuscript.